# UnCoVAEr: Estimating Causal Concept Effects under Visual Latent Confounding

## Abstract

Estimating the effect of human-interpretable concepts on model predictions is crucial for explaining and auditing machine learning systems, as well as for mitigating their reliance on spurious correlations. Most existing approaches assume complete concept annotations, but in practice some concepts may remain unobserved and act as confounders, biasing causal effect estimates. We introduce **UnCoVAEr** (Unobserved Confounding Variational AutoEncoder), a latent-variable model that partitions image latent representations into confounder-related and non-confounding residual components. This allows us to (i) identify which observed concepts are confounded, (ii) obtain corrected unbiased effect estimates via backdoor adjustment, and (iii) learn confounder-proxy variables that align with underlying latent factors. On a controlled semi-synthetic MorphoMNIST benchmark, we show that UnCoVAEr yields substantially less biased effect estimates than prior methods, providing practitioners with a practical tool for trustworthy concept-level causal inference in partially annotated image datasets.

## 1 Introduction

Human-interpretable visual concepts are being increasingly used to explain, audit and control the behavior of machine learning models. Concept-based explanations enable domain experts to ask and answer targeted causal questions such as "how much does hippocampal atrophy, as seen on an MRI, contribute to an Alzheimer's diagnosis?" (Castro et al., 2020) or "which facial attributes drive perceived attractiveness in our annotated dataset?" (Lingenfelter et al., 2022). In practice, however, causal statements at the concept level are only as valid as the assumptions that underlie them. Most existing concept effect estimators implicitly assume that we have measured all relevant visual factors that confound concept-label relations. When important factors are missing from the annotation set, naive observational estimates can be severely biased and lead practitioners to mistaken conclusions and harmful interventions.

Consider a scenario in medical imaging where an interpretable concept-based model concludes that hippocampal atrophy is a dominant predictor of Alzheimer's disease. However, this association may be inflated by confounders such as scanner hardware or acquisition protocols: different scanners alter image appearance in ways that affect how atrophy is manifested and measured, while also correlating with hospital site and diagnostic practices that influence the diagnosis. As a result, a naive estimate of the effect of hippocampal atrophy on diagnosis may capture site- or device-specific artifacts rather than a genuine biological causal effect, potentially misleading auditors and downstream clinical decisions. Next, consider a facial-attribute dataset where a set of annotated concepts such as smiling, makeup or age are used to predict attractiveness. Yet, unannotated factors like skin tone, lighting, or demographic imbalance in the dataset can act as confounders, inflating the estimated influence of certain attributes and masking annotator prejudice (Lingenfelter et al., 2022). Obtaining unbiased causal effect estimates enables practitioners and researchers to estimate bias in datasets (Madras et al., 2019; Di Stefano et al., 2020), decide if they need to collect additional metadata or reweight training examples (Zhao et al., 2023). Moreover, obtaining corrected causal effect estimates can be useful to improve the performance of domain generalization methods by penalizing reliance on spurious attributes (Kumar et al., 2023).

Concept-based models are not new: they can be understood as a principled evolution of feature engineering in which model decisions are expressed in terms of semantically meaningful factors

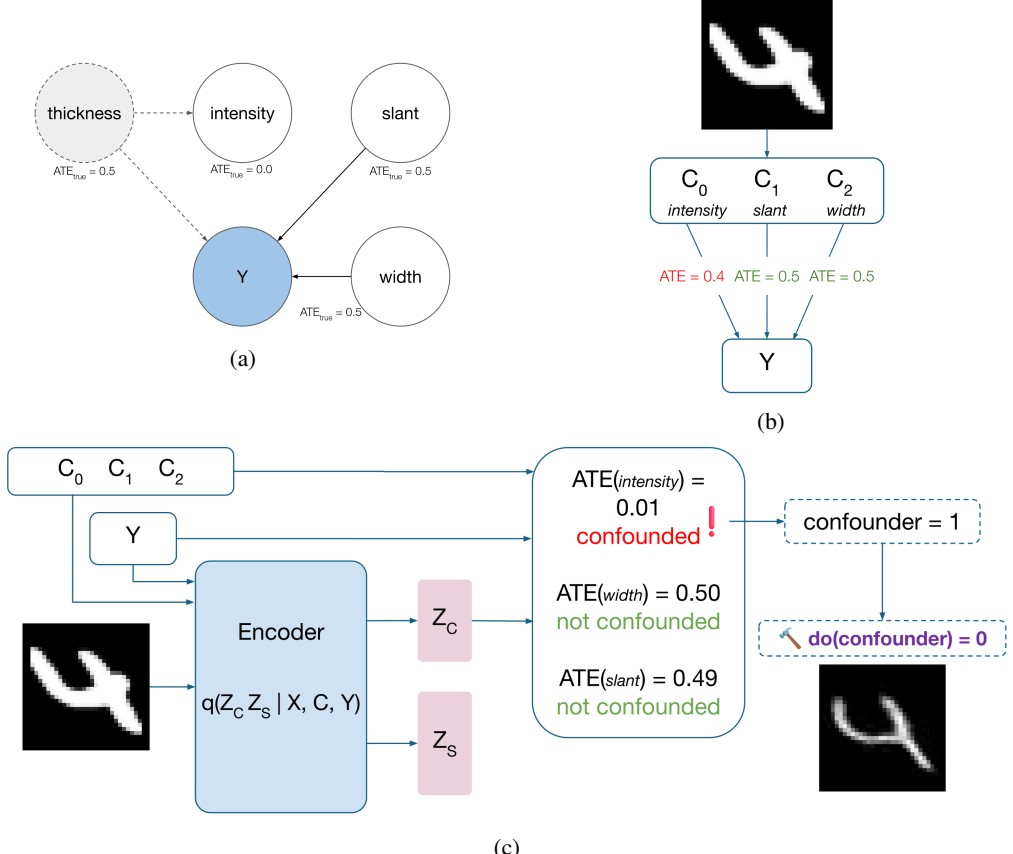

Figure 1: **(a)** The causal graph shows a case where one of our observed concepts $C$ (*intensity*) and our label of interest $Y$ are caused by an unobserved confounder (*thickness*). **(b)** As a result, when computing the effect of *intensity* on $Y$ without taking this confounding into account , we get a wrong estimate. **(c)** Our method *Unobserved Confounding Variational AutoEncoder (UnCoVAEr)* estimates the correct causal effects and finds which observed concepts are confounded. After training our model, we provide $C$, $Y$, and the image $X$ to the encoder, which outputs a confounder-related latent $Z_C$ and a non-confounding residual $Z_S$ used only for image reconstruction. We then perform backdoor-adjustment using the learned proxy $Z_C$ to debias the Average Treatment Effect (ATE) estimation. Additionally, we can intervene on the confounder proxy and interpret its effect.

rather than opaque input dimensions. Early work assumed a fixed, complete set of predefined concepts (Koh et al., 2020); subsequent methods allow concepts to be learned post-hoc or discovered from images (Yuksekgonul et al., 2023; Oikarinen et al., 2023; Sawada & Nakamura, 2022; Shang et al., 2024; Rao et al., 2024), and a growing body of research takes a causal formulation on concept models Dominici et al. (2025b;a); Moreira et al. (2024). Separately, a line of work that builds on proximal causal inference (Tchetgen et al., 2020) studies causal question in the presence of latent confounders that manifest through proxies, estimating treatment effects originally in tabular settings (Louizos et al., 2017; Wu & Fukumizu, 2022; Zhang et al., 2020; Miao et al., 2018; Wang & Blei, 2021) and extending to high-dimensional data such as images (Kaddour et al., 2021; Kompa et al., 2022; Israel et al., 2023; Jerzak et al., 2023; Schulte et al., 2025). Our work sits at the intersection of these two threads: we incorporate a deep latent-variable method from the proximal causal inference literature to robustly estimate causal quantities when unobserved visual concepts confound both observed concepts and the label.

We propose **UnCoVAEr**, a latent-variable model inspired by Causal Effect Variational AutoEncoder (CEVAE) that explicitly accounts for concept incompleteness by partitioning the latent space into two parts: a confounder-related component that explains variation shared between concepts and label, and a non-confounding block that captures residual image variation. This structured decom-

position lets us (i) recover proxies for the confounders, (ii) use them to estimate concept effects via backdoor-adjustment, and (iii) identify which observed concepts are substantially confounded by latent visual factors. We validate our approach on a controlled semi-synthetic Morpho-MNIST benchmark. UnCoVAEr reduces bias in concept-effect estimates compared to prior concept-based and latent-variable baselines.

Our contributions are as follows: (1) we formalize concept incompleteness as a latent confounding problem in image datasets and introduce partitioned-latent representations as an effective inductive bias; (2) we propose a principled criterion to distinguish confounded from unconfounded concepts and correct their effect estimates via backdoor adjustment; and (3) we provide empirical evidence that UnCoVAEr reduces bias in causal effect estimates compared to strong baselines and learns proxy variables that correlate with underlying latent confounders on a controlled semi-synthetic MorphoMNIST dataset.

## 2 RELATED WORK

**Latent-variable proximal causal inference**  UnCoVAEr builds on a line of work that utilizes deep latent-variable models to estimate causal quantities like ATE in the presence of unobserved confounders. CEVAE (Louizos et al., 2017) assumes a causal graph where latent confounders are also causes of observed proxies and uses a Variational AutoEncoder (VAE) formulation to model the data-generating process and estimate ATE with backdoor adjustment. While CEVAE has demonstrated promising empirical performance, its reliance on variational inference raises concerns about identifiability. Rissanen & Marttinen (2021) provide an extensive critique, showing analytically and experimentally that CEVAE can fail when the latent space is misspecified or when the data distribution is complex, despite working in simple synthetic setups, while they also provide simple experiments on digit images. Follow-up works provide identification under limited overlap assumptions (Wu & Fukumizu, 2022) and disentangle instrumental, risk, and confounding factors to better isolate causal effects (Zhang et al., 2020), while Madras et al. (2019) utilize CEVAE to improve causal effect estimates between sensitive attributes and outcome in a fairness setting. At the same time, proximal causal inference literature (Tchetgen et al., 2020) and related proxy-variable identification results provide formal conditions (completeness / rank) under which proxies identify causal effects (Miao et al., 2018; Wang & Blei, 2021). A number of recent works tackles causal effect estimation assuming that images or image-derived features act as proxies for latent confounders. Some approaches apply standard adjustment ideas by learning a model that extracts confounding information from the image via propensity score matching (Jerzak et al., 2023) or by extracting image features (Xu et al., 2021; Schulte et al., 2025). Others develop neural methods that directly learn the necessary adjustment functions from high-dimensional proxies (Kompa et al., 2022). Kumar et al. (2023) also use the image directly to perform backdoor adjustment and use the estimated causal effects of the attributes to regularize classifiers for domain generalization.

**Concept-based explanations**  Concept-based explanation methods such as TCAV (Kim et al., 2018) or Concept Bottleneck Model (CBM) (Koh et al., 2020) treat concepts as interpretable primitives for explaining image classification, enabling interventions and attributing predictions directly to concepts. Follow-up work has extended CBMs to incorporate concepts not predefined in the concept set (Yuksekgonul et al., 2023; Oikarinen et al., 2023; Sawada & Nakamura, 2022; Shang et al., 2024; Rao et al., 2024), while also revealing important limitations, such as whether the learned concepts truly correspond to human-understandable semantics or instead capture spurious shortcuts (Mahinpei et al., 2021; Margeloiu et al., 2021; Havasi et al., 2022). In this line of work, Bahadori & Heckerman (2021) address biases in concept-based explanations arising from confounding information. They propose a two-stage regression technique, inspired by instrumental variable methods, to remove the impact of confounders and noise. Their approach also considers the completeness of the concept set (Yeh et al., 2020), demonstrating effectiveness even when the set is incomplete. Goyal et al. (2020) introduce the notion of *CaCE (Causal Concept Effect)*, defining it as the effect of the presence or absence of a human-interpretable concept on a deep neural network's prediction. They train a conditional VAE to generate counterfactuals by disentangling and intervening on the concept of interest. While they highlight the importance of causality for concept explanations, they rely on the assumption that unobserved confounders do not significantly impact the observed concepts. Gao & Chen (2024) explicitly tackle concept incompleteness by constructing pseudo-concepts or-

thogonal to the observed ones and using a linear predictor to capture residual bias. However, their orthogonality assumption does not allow for confounding.

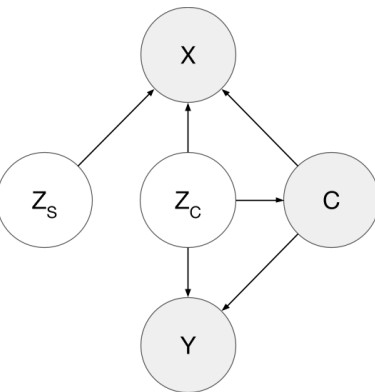

Figure 2: Our causal graph assumption. We assume that the image $X$ is jointly caused by a set of observed concepts $C$, the unobserved confounder $Z_C$ and some $Z_S$ (e.g. writing style, point of view) that is irrelevant for $C$ and the outcome $Y$. $C$ cause $Y$, while the unobserved confounder $Z_C$ causes both $C$ and $Y$.

## 3 PRELIMINARIES AND PROBLEM SETUP

We observe i.i.d. samples $(X, C, Y) \sim \mathcal{D}$, where $X \in \mathcal{X}$ is an image, $C = (C_1, \ldots, C_M) \in \{0, 1\}^M$ are observed binary concept annotations, and $Y \in \{0, 1\}$ is a binary outcome of interest. We are interested in quantifying how changes to a single concept $C_i$ causally affect $Y$. Two common estimands for this are the individual (or conditional) treatment effect (ITE) and the population Average Treatment Effect (ATE):

$$
\begin{aligned}
\mathrm{ITE}_i(x) &:= \mathbb{E}\big[Y \mid do(C_i = 1), X = x\big] \ - \ \mathbb{E}\big[Y \mid do(C_i = 0), X = x\big] \\
\mathrm{ATE}_i &= \mathbb{E}\big[Y \mid do(C_i = 1)\big] \ - \ \mathbb{E}\big[Y \mid do(C_i = 0)\big].
\end{aligned}
\tag{1}
$$

Concept-based explanation methods such as CBMs (Koh et al., 2020) perform *concept-interventions* by editing intermediate concept values and re-evaluating the outcome classifier. These types of interventions and the treatment effects they yield are similar to those obtained by meta-learners in the causal inference literature (Künzel et al., 2019). For example, the S-learner, which is one of the simplest methods for treatment effect estimation, performs the same operation: it fits a predictor of $Y$ given covariates and treatment (concepts in this case) and then estimates treatment effects by changing the treatment value in the input. These estimators coincide with the $do$-intervention only under the ignorability assumption (no unobserved confounding) and can work well enough only in this setting. However, in the presence of unobserved confounders, the estimates will be biased.

**Confounding and backdoor adjustment** Let $V$ denote an observed variable that jointly causes some observed concepts $C$ and the outcome $Y$. If $V$ blocks all backdoor paths from $C_i$ to $Y$, the interventional mean is given by the backdoor formula:

$$
\mathbb{E}[Y \mid do(C_i = c)] \ = \ \int_V \mathbb{E}[Y \mid C_i = c, V = v] \, p(v) \, dv.
\tag{2}
$$

Thus, if $V$ is observed, then Eq. 2 gives an unbiased estimand for the ATE as an immediate consequence of the backdoor-criterion Pearl (1993).

**Proxy learning from images** In our setup, instead of an observed $V$, we have a set of unobserved variables $U$. We assume that the unobserved confounders of interest manifest themselves in the image $X$ (e.g. scanner type, lighting or a separate attribute that affects both annotations and labels). Thus $X$ serves as a high-dimensional proxy for $U$. Because $X$ also contains many features irrelevant to the causal relation between $C$ and $Y$, we aim to learn a lower-dimensional proxy latent $Z_C$ from $(X, C, Y)$ with the following operational properties:

(P1) *Adjustment sufficiency:* conditioning on $Z_C$ blocks the backdoor paths between $C_i$ and $Y$ (so $Z_C$ plays the role of $V$ in Eq. 2);

(P2) *Parsimony:* $Z_C$ is low-dimensional and amenable to downstream estimation and marginalization.

(P3) *Interpretability:* We model $Z_C$ as a binary variable to fit well in our setup of binary concepts and outcomes

We assume the causal structure shown in Figure 2. As we want $Z_C$ to contain only the confounding information for adjustment, we decompose the image into two parts: a discrete confounder-specific latent $Z_C \in \{0,1\}^K$ and a continuous residual latent $Z_S \in \mathbb{R}^d$. Intuitively, $Z_C$ captures the visual variation that (partially) explains both some observed concepts $C$ and the label $Y$, while $Z_S$ contains the remaining image variation that is irrelevant for the causal relationship between $C$ and $Y$, but necessary to model $X$ accurately.

Under (P1), we have:

$$p(Y \mid X, do(C_i = c)) \;=\; \sum_{Z_C} p(Y \mid X, C_i = c, Z_C)\, p(Z_C \mid X)\,. \tag{3}$$

Thus, to estimate $\text{ITE}_i(x) = \mathbb{E}\big[Y \mid X = x, do(C_i = c)\big]$ and $\text{ATE}_i = \mathbb{E}\big[ITE_i(x)\big]$ we need to learn the conditional $p(Y \mid X, C, Z_C)$[1] and the posterior $p(Z_C \mid X)$.

**Identification requirements and limitations**  Identification of causal effects from observational $(X, C, Y)$ rests on standard proxy-type and overlap assumptions. First, the observed image $X$ must carry *sufficient proxy signal* of the unobserved confounder: if the confounder leaves no detectable trace in pixels, then no method can recover its effect. Second, the support of relevant predictive distributions must overlap (positivity), so that the requisite conditional expectations are well defined. Additionally, we assume that there are no *unobserved colliders*: no latent variables caused by both the outcome $Y$ and a concept $C_i$. If such a collider exists, conditioning on it opens a spurious path. Under these conditions, and assuming a sufficiently expressive latent-variable model, our adjusted estimator using $Z_C$ is consistent in principle. We offer three clarifications: (i) we do not require recovery of the true confounder, only that the learned proxy $Z_C$ suffices for valid adjustment; (ii) if the confounder leaves no observable imprint, identification is impossible for *any* method; and (iii) while practical estimation is subject to approximation error, our results show that UnCoVAEr recovers unbiased ATEs whenever these identification assumptions hold.

## 4 UNCOVAER: UNOBSERVED CONFOUNDING VARIATIONAL AUTOENCODER

We now introduce *UnCoVAEr*, a variational autoencoder designed to recover causal concept effects under unobserved confounding.

**Generative model**  The assumed causal graph of Figure 2 leads to the following factorization of the joint distribution:

$$p(X, C, Y, Z_C, Z_S) = p(Z_C)\, p(Z_S)\, p(C \mid Z_C)\, p(X \mid C, Z_C, Z_S)\, p(Y \mid C, Z_C). \tag{4}$$

Our model parameterizes three decoders: $p_{\theta_x}(X \mid C, Z_C, Z_S)$, $p_{\theta_c}(C \mid Z_C)$, and $p_{\theta_y}(Y \mid C, Z_C)$. Because the exact posterior $p(Z_C, Z_S \mid X, C, Y)$ is intractable, we introduce a variational encoder:

$$q_{\phi_c, \phi_s}(Z_C, Z_S \mid X, C, Y) = q_{\phi_c}(Z_C \mid X, C, Y)\, q_{\phi_s}(Z_S \mid X, C, Y).$$

We implement a shared backbone with separate output heads: logits for the discrete confounder-related latent $Z_C$ (parameters $\phi_c$) and Gaussian parameters ($\mu_{\phi_s}, \sigma_{\phi_s}$) for the continuous residual latent $Z_S$. $Z_C$ is sampled with the Gumbel–Softmax relaxation during training (Jang et al., 2017). To reduce information leakage between the two blocks, we additionally include a mutual-information regularizer that minimizes $\text{MI}(Z_C, Z_S)$ using the CLUB estimator with parameters $\psi$ (Cheng et al., 2020). This encourages $Z_C$ to capture confounder-related variation distinct from the residual information in $Z_S$.

---

[1]Note that under the assumed causal graph of Figure 2 $Y$ is independent of $X$ given $C$ and $Z_C$, so $p(Y \mid C, Z_C)$ suffices.

**Training objective**  We maximize the evidence lower bound (ELBO):

$$\mathcal{L}_{\mathrm{ELBO}} = \mathbb{E}_{q_{\phi_c,\phi_s}(Z_C,Z_S|X,C,Y)}\big[\log p_{\theta_x}(X \mid C, Z_C, Z_S) + \log p_{\theta_c}(C \mid Z_C) + \log p_{\theta_y}(Y \mid C, Z_C)\big]$$
$$- \mathrm{KL}\big(q_{\phi_c}(Z_C \mid X, C, Y) \,\|\, p(Z_C)\big) - \mathrm{KL}\big(q_{\phi_s}(Z_S \mid X, C, Y) \,\|\, p(Z_S)\big), \qquad (5)$$

using the following priors: $p(Z_C) = \prod_{j=1}^{K} \mathrm{Bern}(\pi = 0.5)$ and $p(Z_S) = \prod_{j=1}^{d} \mathcal{N}\big(Z_{S_j} \mid 0, 1\big)$.

We augment $\mathcal{L}_{\mathrm{ELBO}}$ with two auxiliary discriminative losses, implemented as small classification heads (following Louizos et al. (2017)):

$$\mathcal{L}_{\mathrm{aux},C} = -\mathbb{E}_{p_{\mathrm{data}}(x,c)}\big[\log q_{\xi_C}(C \mid X)\big], \quad \mathcal{L}_{\mathrm{aux},Y} = -\mathbb{E}_{p_{\mathrm{data}}(x,c,y)}\big[\log q_{\xi_Y}(Y \mid X, C)\big].$$

The auxiliary losses serve two roles: (i) they are used during inference, providing predictors for $C$ and $Y$, and (ii) they encourage representations that capture task-relevant information, sharpening the posterior and improving the quality of learned $Z_C$ as a confounder proxy. Adding the CLUB-based mutual-information estimate $\mathcal{L}_{\mathrm{MI}} = \widehat{\mathrm{MI}}_\psi(Z_C, Z_S)$, The overall training objective is therefore

$$\mathcal{L}_{\mathrm{train}} = -\mathcal{L}_{\mathrm{ELBO}} + \lambda_C \mathcal{L}_{\mathrm{aux},C} + \lambda_Y \mathcal{L}_{\mathrm{aux},Y} + \lambda_{\mathrm{MI}} \mathcal{L}_{\mathrm{MI}}, \qquad (6)$$

where $\lambda_C, \lambda_Y, \lambda_{\mathrm{MI}} \geq 0$ balance the auxiliary and independence terms. In our experiments, setting $(\lambda_C, \lambda_Y, \lambda_{\mathrm{MI}}) = (1.0, 1.0, 0.1)$ yielded the best performance.

The auxiliary $q_{\xi_C}(C \mid X)$ is analogous to the concept layer in concept-bottleneck models (it provides an image-to-concept mapping), while $q_{\xi_Y}(Y \mid X, C)$ functions similar to an outcome layer with a residual connection (Yuksekgonul et al., 2023).

While the original CEVAE utilizes a TARNET-style architecture (Shalit et al., 2017) that fits separate outcome heads per treatment, our model shares decoders and conditions on $C$, since the networks would scale exponentially with the number of concepts. In our experiments we also explore a variant that allocates an independent discrete latent $Z_{C_i}$ for each concept $C_i$ (i.e., separate confounder proxies per concept). This allows us to estimate confounder proxies separately per concept and better interpret their relation with the observed proxies.

We use KL-annealing for the latent KL terms (gradually increasing their weight from 0 to 1 during early epochs) to avoid posterior collapse (Bowman et al., 2016) and temperature annealing for the Gumbel-Softmax relaxation of $Z_C$ (start at $\tau_0$ and reduce to $\tau_{\min}$) to transition from smooth relaxation to near-discrete samples (Jang et al., 2017).

**ATE estimation**  After training, we estimate interventional means by marginalizing over the aggregated posterior of the confounder-latent $Z_C$. Concretely, for each test image $x$ we draw samples $(c, y, z)$ with $c \sim q_{\xi_C}(C \mid x)$, $y \sim q_{\xi_Y}(Y \mid x, c)$, and $z \sim q_{\phi_c}(Z_C \mid x, c, y)$. This yields approximate draws from $q_{\phi_c}(Z_C \mid x)$, analogous to the marginalization strategy in Louizos et al. (2017). For each target concept $C_i$, we intervene by setting it to $c \in \{0, 1\}$ while leaving the remaining concepts $C_{-i}$ at their sampled values, and evaluate

$$\widehat{\mathbb{E}}\big[Y \mid do(C_i = c)\big] \approx \frac{1}{N} \sum_{n=1}^{N} \frac{1}{M} \sum_{m=1}^{M} \mathbb{E}_{\theta_y}\Big[Y \mid C_i = c,\ C_{-i} = c_{-i}^{(m)},\ Z_C = z^{(m)}\Big], \qquad (7)$$

where $M$ (100 in our experiments) is the number of posterior samples per image. For each $C_i$, the ATE is the difference in predicted outcomes under interventions $C_i = 1$ and $C_i = 0$. To detect confounding, we compare the above ATE with the estimated difference in conditional means: $\mathrm{ATE}_{\mathrm{naive}} = \mathbb{E}[Y \mid C_i = 1] - \mathbb{E}[Y \mid C_i = 0]$, which would . We flag a concept as confounded when the computed ATE significantly and systematically differs from $\mathrm{ATE}_{\mathrm{naive}}$. For this, we employ a bootstrap test, in which we resample and recompute ATEs per batch and flag a concept as confounded if the 95% confidence intervals of the ATE do not overlap.

## 5 EXPERIMENTAL SETUP

### 5.1 DATASET

We evaluate UnCoVAEr on a controlled semi-synthetic benchmark derived from Morpho-MNIST (Castro et al., 2019), where digit images are systematically modified along interpretable morphological axes. All experiments use 5 random seeds; for each seed we select a different digit class (0–4).

This design minimizes variation due to digit identity and isolates causal effects arising purely from morphology.

We focus on four pixel-level morphological attributes as binary concepts: *thickness*, *intensity*, *slant*, and *width*. Continuous values for each concept are sampled conditionally from $\mathcal{N}(0.25, 0.01)$ when $C_i = 0$ and $\mathcal{N}(0.75, 0.01)$ when $C_i = 1$. The values are then scaled according to the attribute.

The outcome $Y$ is a synthetic label constructed as a logical rule over the concepts:

$$Y = \mathbf{1}\{\text{thickness} + \text{slant} + \text{width} \geq 2\},$$

i.e., $Y = 1$ if at least two of these three concepts are active, with *intensity* not causing $Y$.

We design three dataset variants to probe distinct confounding patterns:

(i) **Single confounder:** observed concepts: {*intensity, slant, width*}; unobserved: {*thickness*}. Thickness causally influences intensity, making it the only observed concept affected by an unobserved confounder.

(ii) **Common confounder:** observed: {emphintensity, slant, width}; unobserved: {*thickness*}. Here thickness jointly drives both intensity and slant, acting as a shared confounder across multiple observed concepts.

(iii) **Multiple confounders:** observed: {*intensity, width*}; unobserved: {*thickness, slant*}. Both thickness and slant affect intensity through a non-linear XOR causal mechanism, so a single observed concept is influenced by two distinct unobserved confounders.

Across all variants we control the *confounding strength* $\alpha$. For a causal link $C_i \rightarrow C_j$, the label of $C_j$ is set equal to $C_i$ with probability $\alpha$. We evaluate under two regimes: an *in-distribution* (ID) test set with strong confounding ($\alpha = 0.9$) and an *out-of-distribution* (OOD) test set with much weaker confounding ($\alpha = 0.6$), enabling us to assess robustness of ATE estimation under shifts in the confounding mechanism.

Lastly, to ensure that our method correctly adjusts for observed confounders, we construct an additional experimental setup by modifying the *Multiple confounders* variant. We assume that thickness is now observed (slant remains unobserved).

## 5.2 BASELINES AND ABLATIONS

We benchmark UnCoVAEr against latent-variable, concept-based, and feature-adjustment methods.

**CEVAE** (Louizos et al., 2017) is adapted for the image domain with convolutional encoders/decoders. Its difference from our method is that it does not partition the latent space but uses a single continuous latent.

**CaCE** (Goyal et al., 2020) estimates causal concept effects via counterfactual generation. Its original formulation refers to estimating effect on a classifier, rather than the true causal effect. For fair comparison we use the same architecture for encoder/decoder and we train an auxiliary predictor $q_\xi(Y \mid X)$, which we use to assess change in outcome.

**Image-adjustment** (Jerzak et al., 2023) conditions directly on image embeddings by fitting a propensity score model $\hat{e}_i(x) \approx p(C_i = 1 \mid X = x)$ and applying inverse-probability weighting (IPW) to estimate $E[Y \mid do(C_i)]$.

**Concept Bottleneck Model (CBM)** (Koh et al., 2020) predicts $Y$ through an intermediate concept layer $\hat{C} = f(X)$ and enables interventions by editing $\hat{C}_i$.

**Residual CBM (Res-CBM)** augments standard CBM by explicitly modeling variation unexplained by observed concepts. Predictions are of the form $\hat{Y} = g(\hat{C}, r(X))$, where $r(X)$ is a residual representation. During training the concept layer remains fixed and we discretize $r(X)$ with Gumbel-Softmax. To estimate causal effects, we use IPW with $\hat{e}_i(C_i \mid r(X))$.

Finally, we include two meta-learners in the style of S-learners: (i) a **Naive Estimator**, which conditions only on $C$ (biased under unobserved confounding); and (ii) an **Oracle Estimator**, which additionally conditions on the true latent confounder(s), providing an empirical upper bound.

To assess the contribution of each component of UnCoVAEr, we perform the following ablations: (i) removing the image reconstruction term $p_{\theta_x}(X \mid C, Z_C, Z_S)$, (ii) using only a shared discrete latent $Z_C$, (iii) the default model with shared $Z_C$ and residual $Z_S$; and (iv) a variant with separate per-concept confounder proxies $Z_{C_i}$.

# 6 RESULTS

Table 1 reports ATE estimation error across methods, datasets, and test regimes. Several consistent patterns emerge. First, in the *single confounder* setting, UnCoVAEr substantially outperforms all baselines, apart from the oracle which has access to the true confounder. The closest competitor is CEVAE, which itself can be seen as a restricted instance of our model using only a continuous latent.

Second, in the *common confounder* scenario, UnCoVAEr again improves upon feature-adjustment and CBM-based approaches. Interestingly, CaCE performs competitively here. Counterfactual concept editing remains effective when a single latent factor drives multiple observed pathways. Nevertheless, UnCoVAEr maintains strong performance, especially in-distribution.

Third, the *multiple confounders* variant exposes an interesting case. Since intensity is caused by the logical XOR of two latent factors, naive conditioning and CBMs manage to directly learn and exploit the *intensity-Y* relation without accounting for the unobserved confounder at all, performing unexpectedly close to the oracle. Image-based methods, by contrast, are misled by this non-linear dependence. Among them, UnCoVAEr provides the lowest error, though the per-concept $Z_{C_i}$ variant proves unstable in this regime. This suggests that while our structured latent partition is generally robust, learning disentangled proxies remains challenging under interacting confounders.

Finally, across all scenarios, UnCoVAEr shows improved out-of-distribution robustness: errors remain consistently lower than baselines when the strength of confounding shifts from $\alpha = 0.9$ to $\alpha = 0.6$. This supports our central claim that learning an explicit confounder proxy yields more stable causal effect estimates under distributional change.

**Ablations and Diagnostics** Table 1 further shows the effectiveness of the partitioned latent space design of UnCoVAEr. When the same bottleneck latent is used for reconstruction and for recovering the confounder proxy, the method underperforms. Moreover, our hypothesis that the proxy should guide image reconstruction is validated, as is evident in the performance drop. Finally, per-concept latents $Z_{C_i}$ provide marginal gains, but become unstable under complex confounding. Figure 3 indicates that our confounding-detection criterion is generally effective, especially in the common confounder scenario, where it correctly characterizes both observed concepts. We report occasional false positives in the single case and a significant deterioration in the multiple-confounder case, where the naive estimators approximate the true ATE more closely than our estimands.

Table 1: Mean ATE estimation error (MAE, lower is better) across methods, datasets, and test regimes (averaged across concepts). Results are reported as mean ± std over 5 seeds. ID: in-distribution test set ($\alpha = 0.9$); OOD: out-of-distribution test set ($\alpha = 0.6$). Best non-oracle baseline per column is in **bold**.

| Method | Single confounder | | Common confounder | | Multiple confounders | |
|---|---|---|---|---|---|---|
| | ID | OOD | ID | OOD | ID | OOD |
| Naive | $.131 \pm .18$ | $.135 \pm .19$ | $.163 \pm .11$ | $.213 \pm .04$ | $\mathbf{.009 \pm .01}$ | $\mathbf{.009 \pm .01}$ |
| Oracle | $.003 \pm .01$ | $.002 \pm .00$ | $.002 \pm .01$ | $.009 \pm .02$ | $.001 \pm .01$ | $.001 \pm .01$ |
| Image-adjustment | $.168 \pm .16$ | $.133 \pm .18$ | $.440 \pm .24$ | $.183 \pm .14$ | $.109 \pm .14$ | $.117 \pm .15$ |
| CBM | $.136 \pm .18$ | $.136 \pm .18$ | $.163 \pm .11$ | $.214 \pm .04$ | $.011 \pm .01$ | $.012 \pm .01$ |
| Res-CBM | $.331 \pm .20$ | $.418 \pm .09$ | $.171 \pm .17$ | $.560 \pm .40$ | $.287 \pm .24$ | $.253 \pm .23$ |
| CaCE | $.114 \pm .14$ | $.087 \pm .09$ | $.058 \pm .05$ | $\mathbf{.065 \pm .03}$ | $.157 \pm .13$ | $.166 \pm .07$ |
| CEVAE | $.058 \pm .06$ | $.049 \pm .05$ | $.079 \pm .08$ | $.112 \pm .05$ | $.106 \pm .09$ | $.096 \pm .06$ |
| UnCoVAEr (no $p(X)$) | $.113 \pm .07$ | $.112 \pm .06$ | $.064 \pm .06$ | $.119 \pm .07$ | $.210 \pm .14$ | $.209 \pm .14$ |
| UnCoVAEr (only $Z_C$) | $.070 \pm .11$ | $.089 \pm .11$ | $.098 \pm .10$ | $.172 \pm .08$ | $.080 \pm .10$ | $.077 \pm .10$ |
| UnCoVAEr ($Z_S + Z_C$) | $\mathbf{.036 \pm .04}$ | $.040 \pm .03$ | $.055 \pm .07$ | $.097 \pm .06$ | $.070 \pm .05$ | $.065 \pm .05$ |
| UnCoVAEr ($Z_S + Z_{C_i}/C_i$) | $.041 \pm .04$ | $\mathbf{.037 \pm .04}$ | $\mathbf{.047 \pm .03}$ | $.105 \pm .04$ | $.136 \pm .11$ | $.138 \pm .09$ |

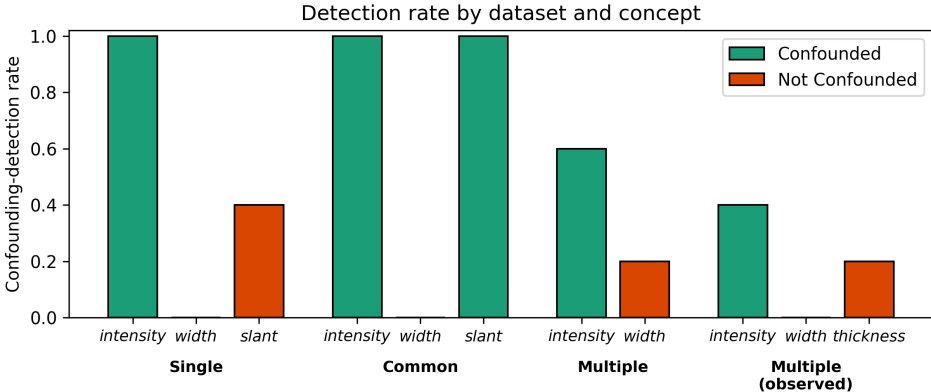

Figure 3: Empirical rate at which each concept was detected as confounded across random seeds (*confounding detection rate*). Results are shown for all three MorphoMNIST variants, plus the multiple-confounders setting where one of the confounders (thickness) is observed.

# 7 DISCUSSION

Our work addresses a critical gap in concept-based model interpretation: the presence of unobserved visual confounders that bias causal effect estimates. While concept-based methods have gained traction for their interpretability, our results demonstrate that ignoring latent confounding can lead to substantially biased conclusions about which concepts truly drive model predictions.

Our experiments reveal interesting nuances in different confounding scenarios. While UnCoVAEr excels with single or shared confounders, performance degrades when confounders interact in non-linear ways (e.g., XOR). In such cases, direct statistical associations remain easier to capture than the underlying more complex causal structure, and all tested image-based methods fail. Handling complex, interacting confounders remains an open challenge requiring further methodological development. Still, the OOD evaluations are encouraging: robustness to shifts in confounding strength indicates that $Z_C$ captures meaningful causal signals rather than overfitting correlations. This robustness is essential for real-world applications where confounding patterns may vary across datasets or deployment contexts.

**Limitations and Future Work** UnCoVAEr's primary limitation is its reliance on the assumption that confounders manifest visually in the image. Our experiments also highlight that complex causal structures or interactions remain challenging for current latent-variable approaches. However, the most critical challenge—and the most important direction for future work—is validating UnCoV-AEr on complex real-world datasets. The true test of our model's practical utility lies in its ability to perform robustly in settings like medical imaging or model auditing, where concepts interact in un-predictable ways and the ground-truth confounding variables are fundamentally unknown. Success-fully demonstrating effectiveness in these noisy, high-stakes environments is essential for moving from a theoretical proof of concept to a reliable tool for causal interpretability in applied domains.

# 8 CONCLUSIONS

We introduced UnCoVAEr, a deep latent-variable model for estimating causal concept effects under visual latent confounding. By partitioning the latent space into confounder-related and residual com-ponents, our method recovers proxy variables that enable valid backdoor adjustment even when key visual concepts remain unannotated. On controlled benchmarks, UnCoVAEr substantially reduces bias in causal effect estimates compared to existing concept-based and latent-variable approaches, while maintaining robustness under distribution shift. Our work highlights a critical considera-tion for practitioners that rely on concept-based explanations: incomplete concept annotations can severely bias causal conclusions. UnCoVAEr provides a practical tool for detecting and correcting such biases, enabling more trustworthy concept-level causal inference in partially annotated image datasets.

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

## A    CODE AND IMPLEMENTATION DETAILS

All code, configuration files, and instructions required to reproduce our experiments are available at `https://anonymous.4open.science/r/causal-residual-concepts-346E/README.md`. The repository includes full implementations of UnCoVAEr and all baselines, as well as scripts for dataset preparation, training, and evaluation. We provide detailed configuration files specifying model architectures, optimizer settings, training schedules, and hyperparameter choices. Additional results, including json files and qualitative figures (e.g., counterfactual visualizations for benchmarked methods), are also included in the repository.

**Reproducibility checklist**

- **Datasets:** MorphoMNIST variants described in Section 5, with generation scripts included in the repository.
- **Evaluation metrics:** mean absolute error (MAE) of ATE estimates, bootstrap uncertainty test for confounding assessment, as described in Section 6.
- **Code availability:** full training/evaluation code and pre-trained model checkpoints are provided.
- **Hyperparameters:** all hyperparameters (learning rate, optimizer type, batch size, KL-annealing schedules, Gumbel-Softmax temperature annealing) are specified in configuration files.
- **Compute:** experiments were run on a single NVIDIA A10 GPU (24GB memory); training a model typically takes around 15 minutes.
- **Randomness:** results are averaged over 5 seeds, with random seeds fixed and logged for reproducibility.

## B    USE OF LARGE LANGUAGE MODELS

Large Language Models (LLMs) were used as assistive tools during the preparation of this paper. Their role was limited to improving readability and presentation: for example, rephrasing paragraphs for smoother academic flow, standardizing LaTeX formatting, and polishing grammar. In some cases, LLMs were also used to suggest more concise ways of summarizing experimental findings. They were not involved in research ideation, experimental design, implementation, or interpretation of results. All scientific contributions are the sole responsibility of the authors, who take full responsibility for the final content.

