# OpenReview forum: "UnCoVAEr: Estimating Causal Concept Effects under Visual Latent Confounding"
_ICLR.cc/2026/Conference — ICLR 2026 Conference Withdrawn Submission_

### Official Review · Reviewer_Njet · 2025-10-22

**Soundness:** 2
**Presentation:** 3
**Contribution:** 2
**Rating:** 2
**Confidence:** 4

**Summary:**

This paper proposes a method called UnCoVAEr to estimate causal effects of interpretable concepts on model outcomes. UnCoVAEr identifies the observed concepts that share a hidden confounder with outcome and estimate causal effect of those concepts by performing backdoor adjustment on the estimated hidden confounder. Experiments are performed on a semi-synthetic dataset.

**Strengths:**

1. The paper is written well and easy to understand.
2. The problem of identifying hidden confounders in image datasets and adjusting them for causal effect estimation is timely and useful to study.

**Weaknesses:**

1. In L259-262, from causal graph in Figure 2, we have $p(Z_C,Z_S\mid X, C, Y) \neq p(Z_C\mid X, C, Y)\times p(Z_S\mid X, C, Y) $ because conditional on $X$, $Z_C, Z_S$ become dependent. But such dependence is not modeled in the variational encoder model.
2. Experiments are performed on only one (semi-synthetic) dataset. This limitation is acknowledged but the paper needs to show results on real-world datasets. Additionally, the experimental setup that the concepts such as thickness/width having causal effect on label is unrealistic.
3. From the results shown in the main table, UnCoVAEr underperforms compared to baselines in three out of six columns. Overall, the current experimental evidence is not enough to appreciate the usefulness of UnCoVAEr.

**Questions:**

See the weaknesses section.

---

### Official Review · Reviewer_kuUE · 2025-10-31

**Soundness:** 2
**Presentation:** 4
**Contribution:** 3
**Rating:** 4
**Confidence:** 4

**Summary:**

The authors propose a method to account for unoberved confounding via proxies in high-dimensional (computer vision) settings. They define a causal graph first, for which their assumptions hold, and introduce further assumptions to make the proxy task allow to properly identify the treatment effects they target for. The DAG is designed to hold for tasks in which we assume that observed concepts cause both the image and the targets, while there are also further unobserved confounders present, causally influencing all of the three variables. They further model task-irrelevant latent features that are unobserved as well. Their proposed method is a VAE where the main ELBO logic borrowed from CEVAE, while they additionally actively disentangle latent space into unobserved latent confounding and task-irrelevant features via CLUB-MI. They verify their model performance on the Morpho-MNIST dataset against other methods, ranging from causal proxy estimation to concept-based models from explainable AI.

**Strengths:**

The paper is very well written. The idea is well-communicated via figures and text equally. The authors contextualize the work well, making it easy for readers and reviewers to evaluate the current state, the relations to older and newer work. The paper is well structured and easy to follow. Argumentation and motivation are clearly articulated. This reviewer further appreciates that all the assumptions are made explicit and logical. While this should be the norm in causality or causally-inspired papers, it unfortunately is not; this makes the paper stand out in this aspect.

The method borrows heavily from CEVAE [1], and can be seen as a neat and important extension of this method. Thus, it justifies its significance. This reviewer generally likes the idea of using proxies when possible, to get better estimations. This work fills in this gap nicely in the field of computer vision.

Finally, the authors submitted their code with this submission, which is highly appreciated and heavily increases credibility in terms of reproducibility.

[1] Causal Effect Inference with Deep Latent-Variable Models

**Weaknesses:**

Although this work shows potential, it does not meet the quality of papers publisehd at ICLR:

Major 1: The results are not convincing. While transparently showing that the proposed methods do not outperform in every category is highly appreciated, the multi-confounding and even common confounder parts raise questions regarding the efficacy of the method or the integrity of the data to a certain degree. This problem is directly related to major issue 2.

Major 2: only a single dataset. The results don't suggest a clear trend on the data. The manuscript would benefit from an evaluation on more datasets, or a detailed exploration why the evaluation has oddities like naive being good at multi-confounding vs. complex methods being bad. The dSprites dataset as used in [1] would be a natural extension. This dataset has even been used not only for instrumental variable regression, but also in proxy tasks, similar to this work, see [2, 3]. Other papers also solved similar evaluation issues with downstream or proxy tasks. In this case, it would have been to estimate causal effects for robust/unbiased predictions in the field of bias mitigation or domain generalization.

Major 3: While the contextualization of the paper is very good and very relevant papers are cited, some of these are missing as appropriate baselines. It is understandable why the authors chose the CaCE method as a baseline, although it is from a different field (XAI), as they faced similar issues in disentangling the latents for unobserved confounding. However, CBM and Res-CBM seem less suitable. It would have been a much better comparison against other causal or proxy methods. DFPV [3] and NMMR [4] are good candidates for example. Another method to conceptually compare to is [5]. While [5] uses a different causal graph, it can be broken down to the same structure. They further separate the latent space into causally relevant and irrelevant parts, also employing an interesting training / inference procedure using VAEs and predictive models. They especially include robustness and bias mitigation experiments to have a proxy task to validate that their method is focusing more on causally stable features rather than spurious / confounding attributes. Finally, another method that comes to mind also disentangles causally relevant and irrelevant unobserved latents in a combined VAE + diffusion approach in adversarial robustness [6]. This method especially resembles the one of this submission, as they also use the CLUB-MI criterion to disentangle these two latent parts.

Medium: while the code base is provided, a summary of implementation details in the appendix would be highly appreciated, given that hyperparameters, and model/architecture design-choices can impact a ranking of models heavily. After skimming the code and config files, a hyperparameter search or model choices could not be identified, which somewhat limits the significance of the results.

[1] Learning Deep Features in Instrumental Variable Regression, Xu et al.
[2] Deep Learning Methods for Proximal Inference via
Maximum Moment Restriction
[3] Deep Proxy Causal Learning and its Application to
Confounded Bandit Policy Evaluation
[4] Deep Learning Methods for Proximal Inference via
Maximum Moment Restriction
[5] Causal Transportability for Visual Recognition
[6] CausalDiff: Causality-Inspired Disentanglement
via Diffusion Model for Adversarial Defense

**Questions:**

Could the authors provide additional results as suggested in the weaknesses section?

---

### Official Review · Reviewer_S2Mj · 2025-11-03

**Soundness:** 3
**Presentation:** 3
**Contribution:** 3
**Rating:** 6
**Confidence:** 3

**Summary:**

This paper proposes a latent-variable model that explicitly partitions an image's latent representation into two independent parts: a confounder-related component and a non-confounding residual component.The model is trained by maximizing ELBO in conjunction with auxiliary losses and a CLUB-based mutual information estimator. This approach enables the model to estimate the ATE of an arbitrary concept $C_i$ on a label $Y$, thereby reducing bias.In the experiments, the authors evaluated the model's ability to estimate concept ATE. UnCoVAEr demonstrated superior performance across all three different confounder settings evaluated.

**Strengths:**

- This paper is clearly written and easy to understand
- This paper proposes a novel bias-reduction model that achieves unbiased causal effect estimation via the backdoor adjustment.
- The experiments demonstrate UnCoVAEr's effectiveness at ATE estimation across various scenarios.

**Weaknesses:**

- Although the authors suggest the model's potential applicability in medical imaging scenarios, the paper lacks experimental validation on actual medical datasets. The evaluation was restricted to synthetic data.

- The training process is complex and training loss is a weighted sum of multiple components. However, the paper provides no analysis of how these weights were determined, lacks a sensitivity analysis of these hyperparameters, and does not discuss whether this weighting scheme generalizes to other cases.

**Questions:**

- Given the model's  potential for medical imaging, could the authors provide an evaluation on a relevant medical dataset, eg. the X-ray Grading dataset or other real datasets?

- Could the authors provide a parameter sensitivity analysis? Specifically, it would be important to know how sensitive the model (e.g., the loss component weights) is to different datasets.

---

### Official Review · Reviewer_zT3G · 2025-11-03

**Soundness:** 1
**Presentation:** 2
**Contribution:** 2
**Rating:** 2
**Confidence:** 4

**Summary:**

The paper studies the estimation of causal effects, Average Treatment Effect (ATE), and Individual Treatment Effect (ITE), when an image variable is involved in a backdoor setting. Let $X$ denote an image and $C_i$ an observed concept extracted from that image. The goal is to estimate $E[Y \mid do(C_i)]$ (ATE) and $E[Y \mid do(C_i), X]$ (ITE), under the assumption that some unobserved concepts within the image can serve as a backdoor set for the pair $(C_i, Y)$.

To achieve this goal, the paper proposes UnCoVAEr, a VAE-based framework that learns a latent representation $Z_c$ from images, intended to serve as a proxy for the unobserved confounders. And then using the learned $Z_c$ for estimating ATE. The effectiveness of UnCoVAEr is demonstrated on a semi-synthetic dataset derived from Morpho-MNIST.

**Strengths:**

**S1**. The motivation and the worked problem are interesting, essential, and important when estimating ATE with high-dimensional variables.

**S2**. The ablation study of UnCoVAEr explores the design choices from multiple valuable perspectives, and the results are insightful.

**Weaknesses:**

**W1**. The paper lacks theoretical guarantees or discussion on **why UnCoVAEr should be able to learn a valid backdoor representation** $Z_c$.
In other words, it is unclear why the architecture proposed in Section 4 satisfies the causal graph assumptions presented in Figure 2. The intuition seems to be that $Z_c$ captures image concepts correlated with $C_i$, while $Z_s$ captures concepts independent of both $C_i$ and $Z_c$. However, correlation does not imply causation: $Z_s$ may contain **descendant concepts** of $C_i$, and adjusting for such descendants could lead to biased causal effect estimates. Therefore, a more formal theoretical justification is needed to explain why the proposed architecture recovers a latent representation aligned with the causal goal.

**W2**. The design of the encoder \{\phi_c, \phi_s\} for estimating the posterior $q(Z_c, Z_s \mid X, C, Y)$ is confusing. In **line 261**, the variational posterior is factorized as:

$q_{\phi_c, \phi_s}(Z_c, Z_s \mid X, C, Y) = q_{\phi_c}(Z_c \mid X, C, Y)q_{\phi_s}(Z_s \mid X, C, Y)$

First, this factorization appears inconsistent with the independence constraints in Figure 2: $Z_s$ should not be independent of $Z_c$ given $X$, because $X$ is a collider. Second, in the implementation, $\phi_c$ and $\phi_s$ share the same backbone (implying that they are dependent), while mutual information between $Z_c$ and $Z_s$ is minimized via a regularizer. It is unclear whether the factorization in line 261 is encouraged or discouraged during training, and this should be clarified.

**W3**. **The model design for handling multiple observed concepts $C_i$ is not clearly explained**. Although the paper provides a high-level discussion (lines 293–296), it remains unclear how the encoder and decoder are structured when different causal relationships exist among the observed concepts. For example, if $C_j$ is a parent, descendant, or independent of $C_i$, should $C_j$ still be included as input to the encoder and decoder? Should such $C_j$ be included in Equation (7) when estimating ATE for a specific $C_i$?

**W4**. The experimental evaluation is limited in terms of **dataset diversity**. All experiments are conducted on a semi-synthetic Morpho-MNIST dataset. It remains unclear how the method performs on higher-dimensional or more realistic data. At minimum, a semi-synthetic experiment based on another dataset (e.g., CelebA-HQ or a medical imaging dataset) would strengthen the empirical validation.

**W5**. The design of the synthetic **data generation process** is limited. The experiments only consider three settings and involve just four variables. Both the ground-truth generation functions for $C$ and $Y$, and the number of variables, should be more diverse. It would be beneficial to define a family of generation functions and sample from it to create multiple ground-truths. Also, additional variables (e.g., color, background texture) could be incorporated to create richer causal structures.

**W6**. The experiments lack **robustness analysis**. It is important to evaluate how violations of key assumptions (e.g., assumption P1 or the causal structure in Figure 2) affect causal effect estimation. Understanding sensitivity to assumption violations is critical before applying the method in real-world scenarios.

**W7**. Although the paper discusses Individual Treatment Effect (ITE) estimation, there is no corresponding evaluation in the experiments. The experiments focus only on ATE estimation, and it remains unclear whether the proposed method can accurately estimate ITE or how its performance compares to baselines.

**Questions:**

**Q1**. In Figure 3, **the confounding detection rate appears lower when one confounder is observed compared to the case where all confounders are unobserved**. Could the authors provide insights into why this occurs? If this behavior consistently appears across different settings, does it imply that the proposed method should, in some cases, reject or ignore certain observed inputs in order to achieve better performance?

Please refer to the weaknesses section for additional questions.

---

### Note · Authors · 2025-12-01

**Comment:**

We thank the reviewers for their thoughtful and constructive comments. We truly appreciate the time and care put into evaluating our work. We have decided to withdraw the paper and will resubmit after carefully addressing the reviewers’ suggestions.

**Withdrawal Confirmation:**

I have read and agree with the venue's withdrawal policy on behalf of myself and my co-authors.